# Harnessing interpretable novel combination of GloVe embedding with deep CNN-BiLSTM neural network for fake news detection

**Sidra khalid**[1], **Shabana Ramzan**[1], **Muhammad Munwar Iqbal**[2], **Nisrean Thalji**[3], **Ali Raza**[4], **Aseel Smerat**[5,6], **Changgyun Kim**[7*], **Norma Latif Fitriyani**[8], **Muhammad Syafrudin**[8*]

1 Department of Computer Science and IT, Government Sadiq College Women University, Bahawalpur, Pakistan, 2 Department of Computer Science, University of Engineering and Technology Taxila, Taxila, Pakistan, 3 Faculty of Artificial Intelligence, Department of Data Science, Al-Balqa Applied University, As-Salt, Jordan, 4 Department of Precision Medicine, Sungkyunkwan University School of Medicine, Suwon, Republic of Korea, 5 Faculty of Educational Sciences, Al-Ahliyya Amman University, Amman, Jordan, 6 Department of Biosciences, Saveetha School of Engineering, Saveetha Institute of Medical and Technical Sciences, Chennai, India, 7 Department of Electronic and AI System Engineering, Kangwon National University, Samcheok, Republic of Korea, 8 Department of Artificial Intelligence and Data Science, Sejong University, Seoul, Republic of Korea

* tiockdrbs@kangwon.ac.kr (C.K.); udin@sejong.ac.kr (M.S.)

**Data availability statement:** The dataset is available at Kaggle: https://www.kaggle.com/datasets/emineyetm/fake-news-detection-datasets.

## Abstract

The important issue of fake news to society is how it affects how society runs in terms of decision-making and public perception. Hence, this study is a comparative analysis of innovative hybrid deep learning models and embedding techniques focusing on interpretability using eXplainable Artificial Intelligence (XAI) for fake news detection. The popular fake news dataset is used to design and test a collection of state-of-the-art models, such as GloVe with CNN-BiLSTM, FastText-Bi-LSTM, and logistic regression with TF-IDF against the CNN and GloVe with BiLSTM and CNN models. In terms of accuracy, LSTM without FastText shows a performance of 98.33%, whereas GloVe with BiLSTM and CNN shows a 99.63% performance. Local Interpretable Model-Agnostic Explanations (LIME) is used to clarify how the input features make decisions on the high precision of the model. The integration of such state-of-the-art models with XAI is one of the major contributions of the study, which brings high accuracy as well as interpretability. Our study's perspective addresses model performance and user trust in the future, laying the foundation for the practical implementation of reliable fake news detection systems.

## Introduction

Fake news has become a significant social issue today, as it is spreading predominantly through all channels of digital communication and influencing the highest decision-making processes, various narratives, and public opinion [1]. Given that fake news exacerbates the public's lack of trust in the media, the rapid spread of misinformation poses a serious challenge for governments, organizations, and individuals seeking reliable sources of information.

**Funding:** This work was supported by the Technology Innovation Program (RS-2024-00507228, Development of process upgrade technology for AI self-manufacturing in the cement industry) funded By the Ministry of Trade, Industry & Energy (MOTIE, Korea).

**Competing interests:** The authors have declared that no competing interests exist.

This requires providing robust detection systems that are accurate and transparent in their workings, thereby fostering openness and building trust with users. The internet's low cost, ease of use, and speed of information delivery have changed how people engage and communicate. As a result, many more individuals now read and seek news on social media and internet portals instead of traditional newspapers. Although social media is a great source of information, it may also have a negative impact on society by influencing important events. The problem of fake news on the internet has become increasingly widespread, particularly after the 2016 U.S. presidential election.

The proliferation of fake news on the internet has reached a critical stage, as evidenced by its exposure during the 2016 U.S. presidential election, where misinformation demonstrated its ability to sway massive opinion shifts [2]. After the 2016 U.S. presidential election, fake information transmission kept maturing through COVID-19 vaccine conspiracies during the Russia-Ukraine conflict, and deceptive disaster accounts about natural disasters. Multimodal fake news, which combines text with images, audio, and video elements, has become a widespread form of deceptive information since its emergence in recent years. This new version of fake news presents a more compelling and deceptive framework, while simultaneously becoming harder for audiences to detect. The complex environment of fake news detection has become even more challenging to manage due to the emergence of deepfakes alongside other AI-generated media, which renders existing detection systems less effective.

Information that has been manipulated by several propagandists in order to spread influential political and other messages over the network [3]. During the 2019 Indian general election, for instance, some individuals used phoney accounts to disseminate fake information on Facebook and Twitter in an attempt to draw attention. Additionally, numerous individuals who utilize the social network platform create a significant amount of false and amazing information. Certain disseminated material can cause confusion and mistrust among social network members. It can be difficult to identify and detect fake news on social media. The rapid dissemination of fake news has a significant impact on millions of people and their real-world surroundings. The problem of fake news creation on social media platforms is not new. Many businesses and well-known individuals utilize various social media platforms to promote their products and establish their brands. All of these operations influence many users to like and spread such news. Fake news is also disseminated throughout the network as a result of this procedure.

However, the internet has become a perfect environment for the dissemination of fake news, including false information, reviews, ads, rumors, political remarks, satires, and more, due to the growing popularity of online social media. Compared to mainstream media, fake news is now more common and extensively disseminated on social media. The widespread use of fake news to mislead and convince internet users of biased information has become a significant concern for both industry and academia [4]. There is a close relationship between rumor and fake news. The purpose of fake news or disinformation is deliberate. On the other hand, rumors are unverified and dubious information that is disseminated without any intention of misleading [5]. It can be challenging to ascertain the intentions of spreaders on social media platforms. Consequently, any inaccurate or misleading material is usually labeled as disinformation on the Internet. It can be difficult to distinguish between authentic and fraudulent information. However, various strategies have been employed to address this issue. In the context of knowledge verification, different Machine Learning (ML) techniques have been employed to identify fraudulent material circulating online [6].

In comparison to conventional ML techniques, Deep Learning (DL), a recently developed technology in the research community, has demonstrated greater success in identifying fake news. DL is superior to ML in several specific ways, including automated feature extraction,

minimal reliance on data preprocessing, the ability to extract high-dimensional features, and increased accuracy. The availability of data and programming tools has increased the popularity and effectiveness of DL-based techniques [7]. Increasingly, companies are leveraging ML and advanced analytics to address these challenges. Natural language processing (NLP) has opened up a world of possibilities for companies looking to comprehend human emotions through data. This is due to the advancements in artificial intelligence. Text, audio, and video are just a few of the natural and social communication formats that NLP can work with. Throughout the textual collection, text mining has helped identify a number of different and helpful patterns and trends. The strategic use of NLP provides firms with a competitive edge over their rivals in today's market environments [8].

Researchers have explored various strategies [9] in recent times for detecting fake news among traditional ML models. ML practitioners use conventional algorithms, such as Support Vector Machines (SVM), with the help of feature extraction tools. Such detection methods are easily implementable but provide limited engineering features and lack the necessary capacity to understand contextual and semantic linguistic meaningfulness. Contextual modeling occurs automatically in DL models, which include Convolutional Neural Networks (CNNs) and Recurrent Neural Networks (RNNs), as well as transformer models such as BERT and RoBERTa, which achieve superior performance outcomes. Recent studies [10] have investigated combination methods between textual content and visual components, as well as contextual factors, because these approaches demonstrate an improved ability to detect fake news across multiple platforms. GloVe embeddings provide a strong framework by partnering with deep CNN-BiLSTM networks to extract the best attributes from word context sense, leveraging the capabilities of deep neural networks. The deployment of these methods yields the successful extraction of syntactic and semantic elements, while providing an interpretable model that addresses the current difficulties with most DL systems. The existing detection methods face limitations that prevent them from supporting generalization across different scenarios, handling multilingual content, and identifying weak or altered material.

Methods for identifying fake news [11–13] using unidirectional pre-trained word embedding models, the majority of practical and currently available approaches employ context-level characteristics and news content. Using bidirectional pre-trained word embedding models with strong feature extraction capabilities has several applications. In traditional ML models, we employ several effective methods for detecting fake news. Compared to the ML model, the DL model provides classical feature-based approaches that do not require any handwritten features. It directly identifies the most appropriate features on its own. Our manuscript investigates the assembly of intricate and dependable frameworks for identifying counterfeit news through avant-garde artificial neural network designs and embedding techniques. The investigation compares diverse models, like GloVe & BiLSTM, FastText & BiLSTM, plus Logistic Regression (LR) & TF-IDF, using a widely-used fake news dataset. With Explainable AI techniques, notably LIME, the research highlights the importance of both clarity and efficiency. LIME provides detailed explanations of how features affect a model's outcomes, improving user confidence in effective systems.

The research is necessary because it simultaneously involves accuracy and interpretability, which enhances the possibility of adopting reliable fake news detection algorithms in real life. Bidirectional Long Short-Term Memory (BiLSTM) networks enhance the capabilities of traditional LSTMs in processing input sequences in both forward and backward directions, and, most importantly for this topic, provide a more detailed context within which words are understood. FastText, on the other hand, is an embedding method that defines itself by using n-grams of letters to represent words, and thus can widely generalize out-of-vocabulary terms and sub-word information, making it ideal for every text classification task. The

novelty of this study lies in its integration of advanced DL with XAI, resulting in a doubly beneficial approach that combines high detection accuracy with interpretability. By tackling both performance and transparency, this strategy opens the door for the implementation of trustworthy and dependable fake news detection systems in practical settings, thereby aiding in the reduction of the negative consequences of false information.

## Problem statement

Despite significant advancements in fake news detection using DL and NLP, current systems often prioritize accuracy at the expense of interpretability. Moreover, many existing approaches cannot generalize across different languages and platforms. The growing complexity of fake news, particularly through multimodal and AI-generated content such as deepfakes, has further challenged the effectiveness of traditional models. These models often fail to provide transparency, which is essential for establishing user trust in real-world applications.

## Research questions

This work, furthermore, is directed at answering the investigation questions that flow from its research objective:

- Which DL algorithms yield the highest accuracy in the fake news detection field?
- What benefit can be obtained from employing the two embedding techniques, GloVe and FastText, on the effectiveness of the detection models?
- Does the XAI method, LIME, make predictions more interpretable?
- What is the best combination of models as well as embeddings to afford both accuracy and transparency?

## Research contributions

The key contributions of this research are:

- The research explores several advanced fake news detection models, including FastText-BiLSTM and GloVe with CNN-BiLSTM, LSTM without FastText, and LR with TF-IDF, which are benchmarked using various datasets to identify their best and worst attributes.
- The GloVe with CNN-BiLSTM model exhibits excellent state-of-the-art capability that surpasses current standards for fake news detection by achieving high precision with robust performance.
- The integration of XAI methodology incorporates LIME technology to make CNN-BiLSTM decision processes more transparent, thereby increasing trust in prediction outcomes.
- The main innovative aspect of this research is that it develops a framework that maintains excellent detection accuracy while enhancing interpretable models for addressing the dual performance and interpretability requirements in fake news detection systems.
- The approach in this research combines the best features of accuracy and interpretability, whereas other studies have typically focused on one aspect. The existing DL approaches for fake news detection achieve high accuracy metrics at the cost of interpretability. This research distinguishes itself by combining LIME with GloVe and CNN-BiLSTM to deliver high performance with interpretability in fake news detection systems, which aims to establish both effective and trustworthy systems.

- LIME enhances the transparency of CNN-BiLSTM by showing the predictive features while performing its operations within the model. The detection system requires transparency to establish trust in scenarios involving genuine fake news.

The paper is structured as follows: The introduction begins with a discussion of fake news and then reviews the relevant literature in the related work section. This is followed by a Section that describes the proposed methodology. The results, along with discussions, are presented in the next Section. The last section clearly outlines the conclusions drawn and suggests directions for future research.

## Related work

The detection of fake news has received substantial attention from society due to its significant impact on society. Almost all previous research has examined various ML and DL methods that aim to solve this problem, ranging from conventional models such as LR and support vector machines to more sophisticated architectures like CNNs and long short-term memory (LSTM) networks. Along with this, representation in features has reached a level where embedding approaches like GloVe, Word2Vec, and FastText have outperformed in various ways for text classification problems. User confidence and understanding of model predictions need to be improved through XAI techniques, as many models still lack interpretability despite improvements in accuracy. Based on these frameworks, this study aims to investigate the detection of fake news in the context of interpretability and accuracy.

### DL-based methods

This study's findings [14] are based on research into FakeBERT, a new approach to false news detection that combines the BERT model with a DL architecture by constructing three concurrent blocks of one-dimensional CNNs with varying kernel sizes and filter counts. It would be extremely valuable in capturing semantic nuance and long-distance dependence in text, and it achieved outstanding precision (98.90%) on a real dataset of fake news [15] collected during the 2016 U.S. Presidential Election by a huge margin across the latest benchmark findings. However, the research also highlights high resource requirements for exhaustive hyperparameter tuning, as well as potential overfitting in the case of unbalanced datasets, despite the results demonstrating the strength of the tested model. In the future, researchers will work on augmenting the processing capabilities of BERT to improve task performance. The paper by [16] proposes a hybrid BiLSTM model with self-attention layers that accurately identifies fake news, achieving a validation accuracy of 98.65%, which improves upon previous strategies. An unbalanced dataset presents challenges for the model, potentially hindering its performance and generalizability across classes. Additionally, enhancing the model for various uses may be difficult due to the intricacy of hyperparameter adjustment. Despite these limitations, the model has the potential to improve the identification of fake content on social networking sites. This study [17] presents a multitask learning model for detecting fake news that uses Capsule Neural Networks (CapsNet) and BiLSTM to identify news items with 97.96% accuracy. Most models are very efficient, but their drawbacks include overfitting, which can occur due to their complexity, as seen in the case of CapsNet. Additionally, extensive and varied datasets are required to ensure generalizability. Furthermore, word measures biases since either, or all, of the essential features—such as source reliability or context—may not be taken into account.

Recent developments [18,19] in NLP have led to numerous studies on detecting fake news on social media platforms, which have transformed the creation and dissemination of news.

One study categorizes fake news into six classes using an attention-based convolutional bidirectional long short-term memory (AC-BiLSTM) approach. It claims to achieve higher accuracy than existing models, but it lacks the speed at which rumors or misleading information can spread. Another research study similarly detects fake news across Facebook and other sites by utilizing DL algorithms to develop models. The performance surpasses that of state-of-the-art methods in terms of accuracy; interestingly, it utilizes heterogeneous features about user profiles, as well as news content, for its processing and analysis. The complexity of user interactions and the intentional dissemination of misleading information, however, provide challenges. Together, these studies highlight the urgent need for robust detection mechanisms to address the challenges posed by fake news in an increasingly digital and interconnected world.

## ML-based methods

The research by [20] examines the negative effects of fake news on society and its rise in conjunction with the growth of social networks. It has reviewed various techniques for detection and finds that traditional ML models usually perform very poorly with accuracies below 90%. Recent neural network models, on the other hand, have shown notable advancements; some have achieved up to 90.3% accuracy and 97.5% recall, demonstrating the efficacy of sophisticated methods such as N-gram vectorization in precisely identifying fake and authentic news. The assessment underscores the need for ongoing research to enhance detection systems and effectively counter misinformation. To improve classification metrics when paired with LSTM networks [21], the research presents a novel false news detection method that utilizes real data mining for auxiliary features. However, it acknowledges limitations, such as the GloVe vector's inability to adequately capture information from secondary elements like domain names, which do not accurately reflect real language. Furthermore, the reliance on online scraping may introduce unpredictability and inconsistencies in the data acquired, thereby compromising the model's resilience.

One of the studies by [14] achieves remarkable accuracy in detecting fake news by integrating GloVe embeddings with LSTM neural networks. Despite its effectiveness, it has issues with contextual understanding. It relies too heavily on pre-trained models, suggesting that it needs to be improved to better capture all aspects of incorrect information and account for social context. The second research study by [15] on the spread of fake news employed a combination of social media and content analysis features. Still, it failed to measure the impact of echo chambers and the rapid dissemination of phony information across platforms. Both studies emphasize the need to make models more interpretive and flexible and suggest further research on developing stronger frameworks that can accommodate the complexity of false information and its associated issues, thereby making decision-making processes more transparent.

## Interpretable AI-based methods

In the study by [22], a description is given of the OPCNN-false model, which was designed as an optimized Convolutional Neural Network for efficient detection of fake news in social media. The optimal performance of the model was demonstrated using four benchmark datasets and proved to outperform traditional ML methods in terms of accuracy, precision, recall, and F1-measure. The methodology promises to alleviate the social and operational harms of fake news and emphasizes the wisdom in addressing such issues [23,24]. The paper [25] reviews the emergent field of rumor theory and fake news identification, particularly concerning datasets and DL techniques. Discussed are dependencies such as rumor

classification systems, including categorization detection, tracking, and stance, while perusing the premises of existing work. Further research is necessary to fill the identified gaps, particularly regarding the development of user characteristics in combination with DL models. This overall research highlights the importance of new methods in combating fake information on social media.

A total of 6,500 news pieces were collected as part of the research work [26]. In total, eleven different ML algorithms were employed, resulting in an accuracy of 94.47%, with precision, recall, and F1 scores accounting for 95%. Some limitations pertain to the high quality of the datasets, including the exclusion of aspects such as social media involvement, and difficulties in real-time detection. These findings suggest that further research is needed to enhance model performance monitoring in dynamic online environments. The research [27] papers showcased the use of N-gram and TF-IDF approaches with classifiers such as SVM and KNN. They reported an accuracy of approximately 92% but showed weaknesses in languages with complex phenomena. Indeed, Bidirectional LSTM-based DL approaches have proven to deliver better results, as long as the computation and the ability to handle large datasets can be afforded.

### Research gap summary

Despite pre-trained embeddings like GloVe and FastText increasing accuracy, they are not fully utilized in a specified domain language and thus require continuous adaptation in the detection systems as an ongoing process. Abstracting the fake news detection techniques yields an analysis of existing techniques into two parts or categories: network-based and linguistic-based detection methods, as well as hybrid ones that combine both. It also included BiLSTM classifiers and BERT-based detectors, which are backed by techniques like LIME to demonstrate their interpretability to users. While these methods classify fake news quite well, they are relatively not easy to be transparent and trusted because the models are necessarily complex in their development. Additionally, the problem of dependence on specific datasets may also limit generalization across different contexts.

## Proposed methodology

Fig 1 expresses the workflow of the proposed approach. The first step in the proposed framework involves preprocessing the dataset using Synthetic Minority Oversampling Technique (SMOTE), followed by tokenization and padding to adapt the text data for embedding representation. Preparations of embeddings involve the TF-IDF feature extraction in the LR model and the utilization of pre-trained embeddings such as GloVe and FastText in deep-learning models to extract both semantic and contextual features of the text. A total of four models have been implemented and trained: LR with TF-IDF for a traditional ML approach, GloVe with CNN-BiLSTM to combine convolutional feature extraction with sequential learning, Basic LSTM to explore contextual relationships within the text, and FastText-BiLSTM to utilize sub-word-level information for enhanced classification capabilities. Each of these models is evaluated on several performance metrics, including accuracy, precision, recall, and F1 score, to facilitate a thorough comparison across the different models of how well they can distinguish between fake and real news articles.

### Dataset collection

The dataset we use in this research focuses on real and fake news collected from the Fake News Detection dataset [28]. Comprising 44,898 records, it has five attributes: title, text,

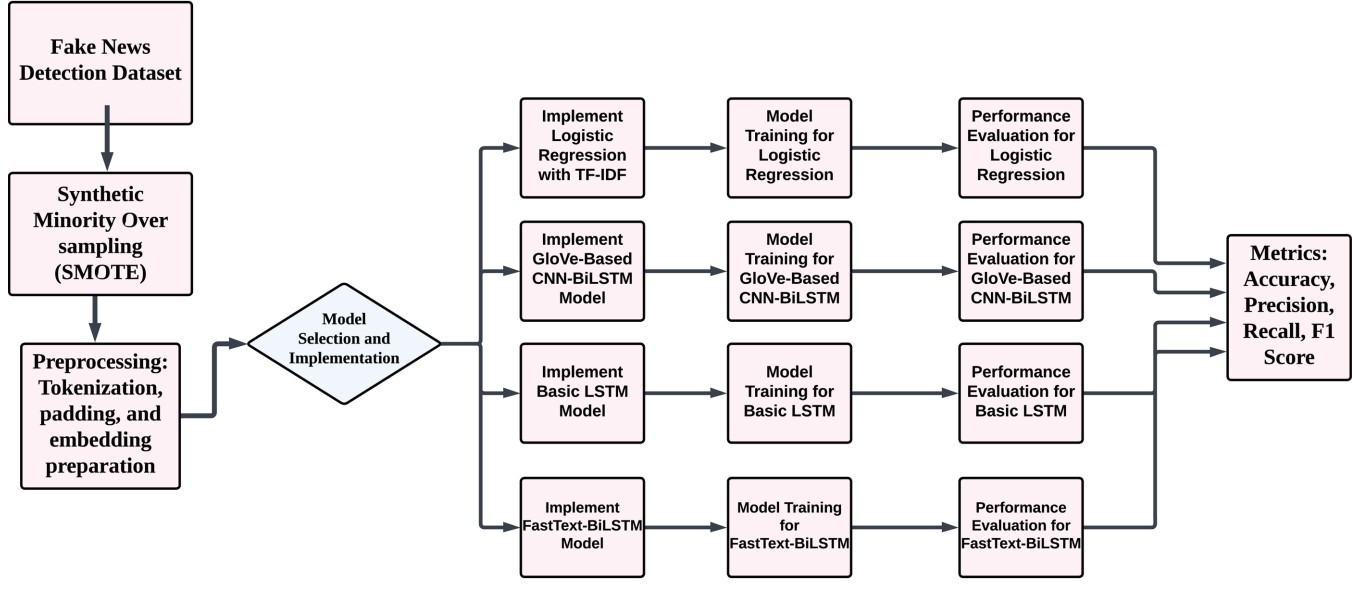

**Fig 1. The proposed methodology overview.**

subject, date, and label. The fake articles were collected from various sources, including PolitiFact and Wikipedia, and the true ones were sourced from the Reuters website. The main topic is world news and politics, which spans the period from 2016 to 2017. The real news articles were gathered from the Reuters news agency, a globally recognized and credible source for authentic journalism. The information collection process combined automated scraping techniques for sorting articles from these sources with manual verification operations to validate the authenticity of the records. These scholarly articles were published between 2016 and 2017, during which the distribution of fake news reached its peak, especially in political environments. The detail of the dataset is given in Table 1. This dataset provides a robust foundation for evaluating and training ML and hybrid models for fake news detection. It has no missing values but a slightly imbalanced dataset, which we address using the SMOTE to ensure the model's performance.

The fake news data were mainly taken from PolitiFact and Wikipedia, and the real news came from Reuters, a famous traditional news outlet worldwide. PolitiFact, despite being trustworthy, may exhibit small biases in its decisions to call out fake content, depending on what journalists consider true or false. Similarly, relying solely on Reuters for news may lead the model to perform well in specific styles or situations, but not necessarily in all cases. We employed preprocessing techniques to guide the model in focusing on language aspects rather than the unique characteristics of each country.

**Table 1. Dataset description.**

| Attributes | Data Type | Class Distribution |
|---|---|---|
| Title | Text (Object) | |
| Text | Text (Object) | Fake News (Label 0): 23,481 articles (52.30%) |
| Subject | Categorical | |
| Date | Date (Object) | Real News (Label 1): 21,417 articles (47.70%) |
| Label | Numeric (INT) | |

## Data preprocessing

Data preprocessing is crucial for preparing the raw text data for embedding generation and input to ML models. The following detailed steps outline the preprocessing methods applied to the dataset:

- Tokenization is performed manually, without the aid of external libraries. It involved breaking a text column down into single words (tokens) by using whitespace and punctuation. Tokenization provides a structured textual representation of raw text, broken down into smaller, analyzable units.
- After tokenization, the sequences of tokens had different lengths. To maintain uniformity and help gain compatibility with DL models, all sequences were padded to a fixed length of 500 tokens. This was done using the pad sequences method, which adds zeroes to the front of shorter sequences. The maximum sequence length (500 tokens) was chosen based on the longest article in the dataset to ensure that no meaningful information was lost due to truncation.

## Data balancing

The class imbalance present in this dataset has shown that the share of fake news instances (label 0) is only 52.30%, while for real news instances (label 1) it is 47.70%. This situation can lead to the model being biased against the majority class during training. To mitigate incomplete data, the SMOTE is employed in the training data. By interpolating specified instances to generate synthetic samples, this technique allows both classes to be equally represented during training. This thereby reduces the biasing of the model and increases its generalization effect by providing the model with the opportunity to learn more from both classes. Fig 2 depicts what happens before and after SMOTE is applied to the data.

In contrast, under-sampling involves the removal of random minority samples, which could waste important data points, while SMOTE enhances class balance by conserving the original samples. Through this technique, the model can acquire equivalent information

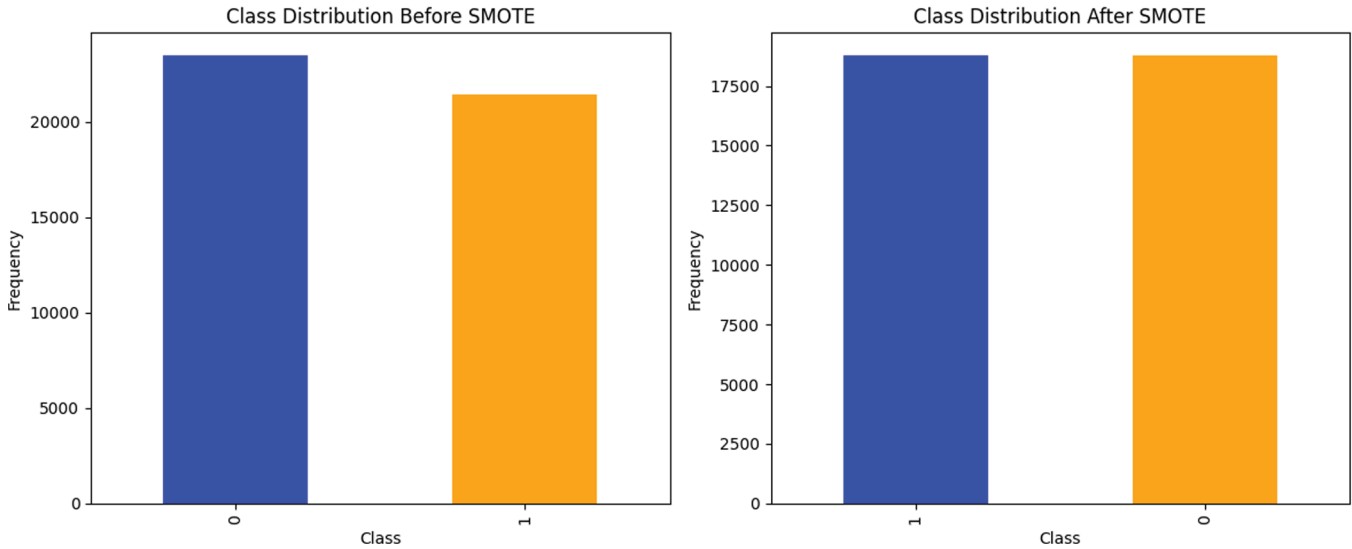

**Fig 2. The analysis before and after SMOTE is applied to the data.**

from the classes without losing important samples. We chose SMOTE over other oversampling methods due to the moderate imbalance in our dataset, as it allowed us to preserve the diversity of training samples.

## Dataset splitting

To assess the effectiveness of the model, the dataset has been partitioned into two separate portions: a training dataset and a testing dataset. The split ratio was kept to 80%-20%, being meant for training and testing, respectively. Under this configuration, the model receives sufficient data exposure for training while reserving a portion of the data for the testing phase to evaluate the model's actual performance. Stratified sampling is used to maintain the original class proportion within both subgroup datasets, thereby preventing the training and testing restrictions from being skewed by class. This encourages both fake and real news to be generally accountable for more real considerations made throughout the training process when evaluating model performance.

## Data validation

Data validation is done to ensure the data is well-formatted and uniform. The tokenized and embedded data, however, have undergone rigorous inspection to ensure that all records have an equal length regarding padding. This is to avoid inconsistent model input since DL models accept input of fixed length. Additionally, the processed dataset contains no missing values or null entries, making it complete and eliminating issues related to model training.

## Feature embedding representations

This research converts textual data into a numerical representation suitable for DL models using pre-trained word embeddings. FastText and GloVe embeddings are utilized for various test purposes to examine their effects on model performance. Mapping tokens with dense numerical vectors enables the model to capture more semantic and contextual relations within the text.

The embedding representation process is illustrated in full detail through Fig 3. This figure demonstrates the tokenization steps that generate numerical input through the pre-trained word embeddings of FastText and GloVe. The diagram illustrates a method for handling out-of-vocabulary words through zero vector assignment, ensuring consistent coverage for every token.

**FastText embeddings.** Pre-trained FastText embedding cc.en.300.bin is downloaded and attached to the model pipeline. The fast text proves very helpful for various text classification tasks in which a word is either a subword or a substring, which means it does not exist in the vocabulary. It has a 300-dimensional dense vector representation for each word in the vocabulary, covering both semantic meaning and morphological features. An embedding matrix is then created by mapping each token in the dataset's vocabulary to its corresponding FastText vector. For tokens not part of the FastText vocabulary, a zero vector has been assigned to them for uniformity and coverage of all tokens. It is now possible to consider all news articles as a series of FastText vectors, replacing each token in the texts with its corresponding vector. It converts the text into a structured numeric input that DL models can use, while preserving the necessary local and contextual details of the text to classify it accurately.

**GloVe embeddings.** In contrast to fast text, we use the GloVe pre-trained embeddings, glove.6B.300d.txt, as a reference. GloVe brings together words in a much broader semantic

**Flowchart of the Embedding Representation Process**

**Fig 3. Flowchart of the embedding process converting news text into vectorized input using FastText or GloVe for DL models.**

context by building word vectors addressing global co-occurrence counts in a large collection. A similar and shared approach is followed for GloVe embeddings, where an embedding matrix is created to map the tokens in the dataset's vocabulary to 300-dimensional vectors. Tokens not found in the GloVe vocabulary will also receive a zero-vector assignment. Input formatting remains the same when using GloVe embeddings, leveraging their semantic richness. An article thus becomes a set of GloVe embeddings, allowing the models to work with global relationships between words instead. This would enhance understanding of the textual data and the larger context provided through Fast Text's sub-word-based embeddings.

## Applied models development

This study focuses on the development and testing of various ML and DL models for unique classifications, specifically differentiating between real and fake news articles. Models may be selected, although those with the power to process text more effectively are those that are

inclined to extract semantic and contextual patterns. The following models have been implemented:

**FastText-BiLSTM model.** The FastText-BiLSTM model leverages the benefits of FastText embeddings and BiLSTM networks for improved classification of text data into either fake or real news. First, it tokenizes the input text and pads it. It is then further represented using pre-trained FastText embeddings, which have sub-word-level features, and these embeddings have a dimensionality of 300. During the training phase, these embeddings have been frozen so as to preserve their pre-trained semantics. It is passed through a BiLSTM layer consisting of 128 units, which enables bidirectional processing of long-term dependencies within the text and captures contextual relationships. Finally, the process culminates in a dense output layer with a sigmoid activation function, which performs binary classification and outputs probabilities to label the input either as fake (label 0) or real news (label 1). The effectiveness of the model in detecting fake news is revealed through the use of performance metrics, such as accuracy, precision, recall, and F1 score, to ensure its appropriateness for such applications. Fig 4 shows the process flow of the FastText-BiLSTM Model. Table 2 contains the FastText-BiLSTM Model Parameters and Values.

## LSTM model

LSTM model without pre-trained embeddings to ascertain its performance on the data. The model structure includes an embedding layer with a vocabulary size derived from the tokenizer's word index, and an embedding dimension of 300, allowing the model to learn embeddings during training [29]. A Bidirectional LSTM layer with 128 units interprets dependence contextually from both forward and backward ways of representing text data. The last layer consists of a dense layer with a sigmoid activation function for binary classification. The model utilizes an Adam optimizer with the binary cross-entropy loss, evaluating performance based on accuracy. The LSTM model trains for five epochs under a batch size of 64 with 20% of the training data for validation. Performance metrics have been computed based on accuracy, with a detailed classification report generated from predictions made on test data.

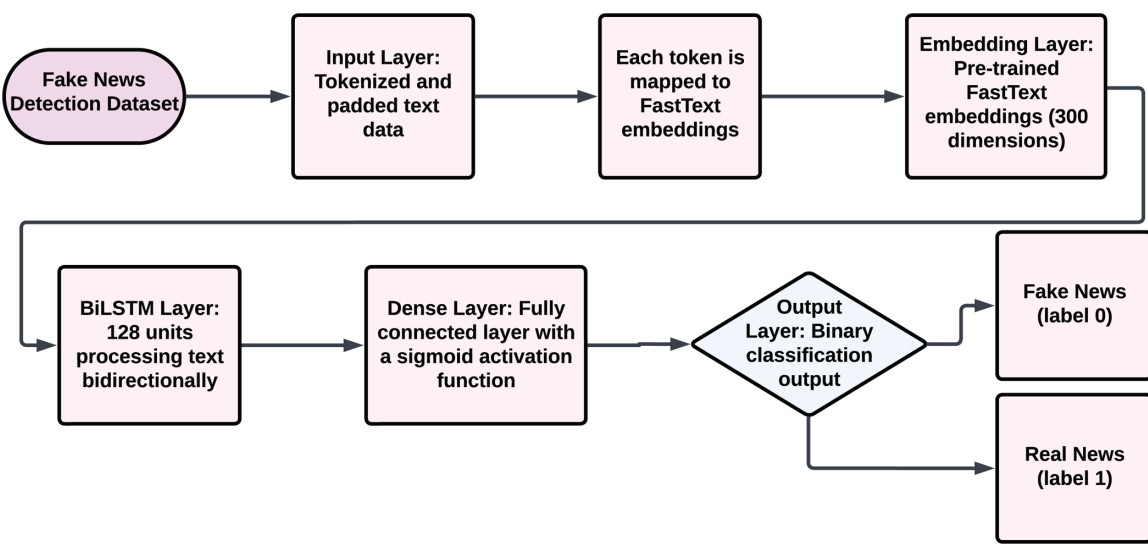

**Fig 4. The process flow of the FastText-BILSTM model.**

**Table 2**. **FastText-BiLSTM model parameters and values.**

| Parameter | Value |
| --- | --- |
| Embedding Input Dim | vocab_size |
| Embedding Output Dim | 300 |
| Embedding Weights | embedding_matrix |
| Embedding Trainable | False |
| Embedding Input Length | 500 |
| LSTM Units | 128 |
| Return Sequences | False |
| Dense Units | 1 |
| Output Activation | sigmoid |
| Optimizer | adam |
| Loss Function | binary_crossentropy |
| Metrics | accuracy |
| Epochs | 5 |
| Batch Size | 64 |
| Validation Split | 20% |

As an additional feature, it caps off the evaluation of the model's effectiveness. Fig 5 shows the process flow of the LSTM Model. Table 3 contains the LSTM Model parameters and values.

**LR with TF-IDF.**   Term Frequency-Inverse Document Frequency (TF-IDF) features are used to train a LR model [30–32]. This traditional ML model serves to evaluate the comparative effectiveness of DL models. Transform the text into a sparse matrix using the TF-IDF method. With this representation, words gain importance in relation to the context in which they are used. Now, apply the LR model to classify TF-IDF transformed text data as fake or real news. An approach used to compare the performance improvements of simpler traditional architectures with their DL counterparts. Fig 6 shows the process flow of LR with TF-IDF. Table 4 contains the LR with TF-IDF Parameters and Values.

## Proposed GloVe-based CNN-BiLSTM model

A hybrid CNN-BiLSTM model leverages pre-trained GloVe embeddings to evaluate its performance in comparison to FastText-based models. The Embedding Layer in this model utilizes the pre-existing GloVe embedding to encode the input text accurately. The CNN layer would then extract local features and patterns from the text. After processing, the final output will be passed on to the BiLSTM, which captures all global dependencies and contextual

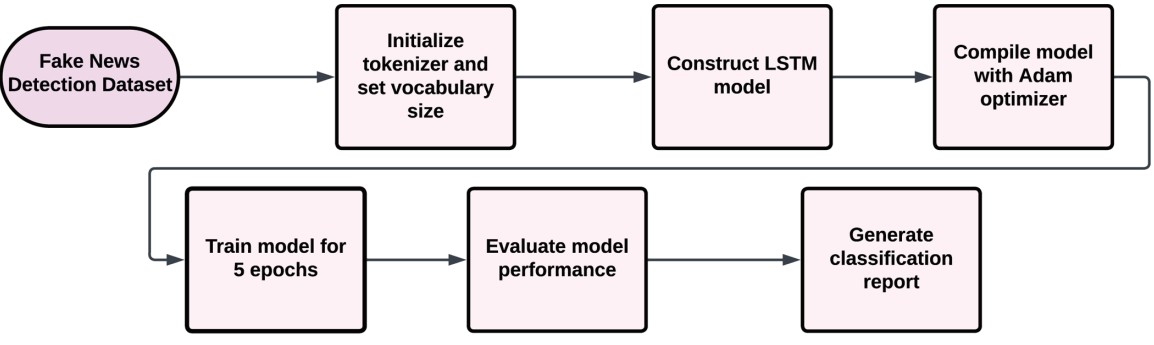

**Fig 5. The process flow of the LSTM model.**

**Table 3**. LSTM model parameter and values.

| Parameter | Values |
|---|---|
| Embedding Input Dim | vocab_size |
| Embedding Output Dim | 300 |
| LSTM Units | 128 |
| LSTM Type | Bidirectional |
| Dense Units | 1 |
| Output Activation | sigmoid |
| Optimizer | adam |
| Loss Function | binary_crossentropy |
| Metrics | accuracy |
| Epochs | 5 |
| Batch Size | 64 |
| Validation Split | 0.2 |

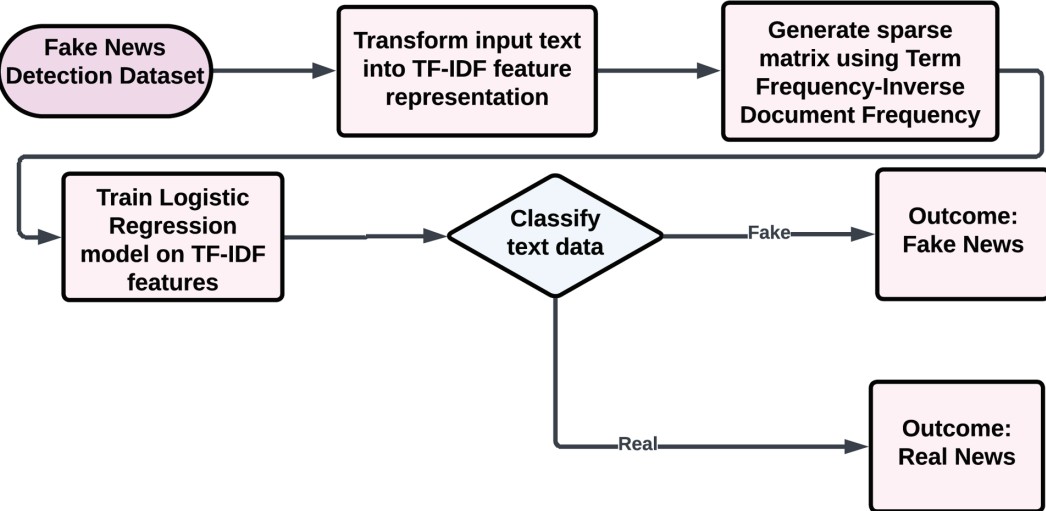

**Fig 6. The process flow of LR with TF-IDF.**

**Table 4**. LR with TF-IDF parameters and values.

| Parameter | Value |
|---|---|
| TF-IDF Max Features | 5000 |
| TF-IDF Representation | X_tfidf |
| Solver | lbfgs (default) |
| Regularization | C=1.0 (default) |
| Loss Function | Log-Loss |

relationships. Subsequently, the output layer would be a dense layer with a sigmoid activation function for classification based on the preferred categories. An integrated system like this combines the advantages of the convolutional portions for feature extraction with the sequential parts for understanding long-term dependencies, together with the semantic richness and contextual relevance imparted by the GloVe embeddings. Fig 7 Process flow of Glove-Based CNN-BILSTM Model. Table 5 contains the GloVe-Based CNN-BiLSTM Model Parameters and Values.

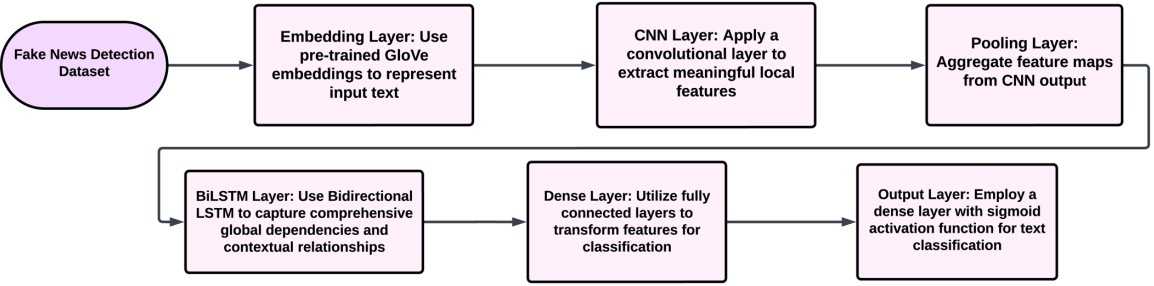

**Fig 7. The process flow of the Glove-based CNN-BILSTM model.**

**Table 5**. GloVe-based CNN-BiLSTM model parameters and values.

| Parameter | Value |
| --- | --- |
| Embedding Input Dim | vocab_size |
| Embedding Output Dim | 300 |
| Embedding Weights | embedding_matrix |
| Embedding Trainable | False |
| Embedding Input Length | 500 |
| Conv1D Filters | 128 |
| Kernel Size | 5 |
| Activation Function (Conv1D) | relu |
| LSTM Units | 100 |
| Dropout Rate | 0.2 |
| Recurrent Dropout | 0.2 |
| Dense Units | 1 |
| Output Activation | sigmoid |
| Optimizer | adam |
| Loss Function | binary_crossentropy |
| Metrics | accuracy |
| Epochs | 5 |
| Batch Size | 64 |
| Validation Split | 20% |

## Application of XAI-based LIME analysis

To enhance the interpretability of the model's predictions and gain insight into how the model makes its decisions, we applied LIME after training the models. LIME helps explain why a model made a particular prediction by approximating it with a simpler, interpretable model near the prediction. LIME was applied to the overall final model after training the LSTM, GloVe + BiLSTM, and FastText + BiLSTM models. Rather than explaining individual models separately, LIME was used to interpret the predictions of the final ensemble or individual predictions after all models were trained and their results were aggregated.

## Hyperparameters selection

In research experiments, the selection of hyperparameters for each model is based on prior studies and iterative testing. For applied DL models, FastText-BiLSTM, GloVe-CNN-BiLSTM, and LSTM, we conducted manual tuning by observing validation accuracy and loss. We performed careful manual tuning using validation performance as a guide. The final selected hyperparameters are summarized in Table 6. The key hyperparameters are chosen due to the following reasons:

**Table 6. Key hyperparameters for DL models.**

| Model | Embedding | Units | Epochs | Batch Size | Optimizer | LR | Val. Split |
|---|---|---|---|---|---|---|---|
| FastText-BiLSTM | FastText (300d) | 128 LSTM | 5 | 64 | Adam | 0.001 | 20% |
| GloVe-CNN-BiLSTM | GloVe (300d) | 128 LSTM + 64 CNN | 5 | 64 | Adam | 0.001 | 20% |
| LSTM | GloVe (300d) | 128 LSTM | 5 | 64 | Adam | 0.001 | 20% |

- **Number of LSTM/CNN units:** 64, 128, and 256 were tested; 128 units yielded the best performance.
- **Embedding dimension:** Fixed at 300 using pre-trained GloVe and FastText embeddings.
- **Batch size:** 32, 64, and 128 were evaluated; 64 led to stable convergence.
- **Epochs:** Capped at 5 based on early stopping to avoid overfitting.
- **Optimizer:** Adam optimizer was chosen for its adaptive learning rate and consistent results.
- **Learning rate:** Default setting of 0.001 was retained as further tuning did not yield significant improvements.
- **Loss function:** Binary Crossentropy for all models due to the binary classification task.
- **Validation split:** 20% of training data was reserved for validation during model tuning.

## Model evaluation metrics

As of now, all of the models can be evaluated according to the following metrics: True Positive (TP) — cases where the model correctly predicts a positive outcome; True Negative (TN) — cases where the model correctly predicts a negative outcome; False Positive (FP) — cases where the model incorrectly predicts a positive outcome; and False Negative (FN) — cases where the model incorrectly predicts a negative outcome. Using these values, we can compute the following performance metrics:

### Accuracy

Overall correctness of the model:

$$Accuracy = \frac{TP + TN}{TP + TN + FP + FN} \tag{1}$$

### Precision

The measurement of the accuracy of positive prediction is called precision:

$$Precision = \frac{TP}{TP + FP} \tag{2}$$

### Recall

Recall (also known as Sensitivity) measures the ability of the applied model to correctly identify all positive instances:

$$Recall = \frac{TP}{TP + FN} \tag{3}$$

### F1 score

The F1 score is the harmonic mean of the recall and precision scores, providing a single metric that balances both measures, especially when the class distribution is imbalanced. It reflects the model's ability to achieve a balance between precision and recall rather than focusing on only one of them.

$$\text{F1-Score} = \frac{2 \times \text{Precision} \times \text{Recall}}{\text{Precision} + \text{Recall}} \qquad (4)$$

### Receiver operating characteristic-area under curve (ROC-AUC)

The ROC-AUC score is a measure of the performance of a classification problem across different threshold settings. The true positive rate is plotted against the false positive Rate to visually assess how well a model separates the two classes.

The metrics above were computed from the test data, intended to measure how well each model identifies fake news.

## Results and discussion

This section presents results based on different models built for the fake news detection task using interpretability through Explainable AI. We test and analyze various DL models, including FastText-BiLSTM, LSTM, GloVe with CNN-BiLSTM, and LR with TF-IDF for performance evaluation to uncover their accuracies along with the way models predict.

### Experiment design

This experiment was conducted on Google Colab to evaluate various ML and DL algorithms for detecting fake news using the Fakes dataset from Kaggle (S1 Code). The dataset was processed to tokenize the text entries, apply lemmatization, and make all sequences of uniform lengths. It is then split into a training component (80%) and a test component (20%) via stratified sampling to ensure balanced classes. The feature representations include TF-IDF for traditional models and pre-trained embeddings like FastText and GloVe for deep-learning models. As a baseline, training using LR, supervised by TF-IDF, LSTM, CNN-BiLSTM with GloVe, and BiLSTM with FastText, was employed to demonstrate performance comparisons. Accuracy, precision, recall, F1 score, and ROC-AUC were the performance metrics for the evaluation. Using the LIME tool, model prediction interpretations were made to better understand the importance of features and improve explainability.

The dataset followed stratified sampling to divide itself into (80%) training data and (20%) testing data. These particular ratios were selected to achieve an adequate quantity of training data, along with a reliable and unbiased evaluation subset. Using a split of (80%)-(20%) in ML experiments is standard practice when working with datasets containing 44,000 records and above, since it provides optimal training and testing performance. A (20%) test set was utilized throughout the experiment, running on Google Colab, because it decreased resource usage and computation time, yet upheld evaluation standards.

### Model performance analysis

The study evaluated different models in the fake news detection dataset by using classification accuracy, precision, recall, and F1-score. It revealed insightful information regarding how well a model can classify articles as fake or real and the performance of hybrid DL architectures with pre-trained word embeddings.

**Overall performance metrics results.**  The assessment evaluates the models' ability to classify news articles effectively, and the results validate the hybrid models, particularly those combining pre-trained embeddings (such as GloVe and FastText) with BiLSTM or CNN layers, which outperform their classic ML counterparts, such as LR, significantly. Among the models tested, the hybrid GloVe + CNN-BiLSTM achieved the highest accuracy of 99.63%. The next hybrid of LR with TF-IDF is very close behind with an accuracy of 98.79%. The LSTM without FastText achieves a decent accuracy of 98.33%, indicating the effectiveness of this model. On the other hand, BiLSTM with an accuracy of 95.53% using FastText was good but less than these hybrid high-accuracy models. This fact essentially highlights that hybrid models utilizing advanced embeddings and DL layers outperform. Table 7 presents a comparison of the model's performance.

**BiLSTM + FastText.**  This FastText-BiLSTM model achieves an accuracy of 95.53%, demonstrating its effectiveness in handling out-of-vocabulary words through sub-word representations. It is below the GloVe-based model, sharing much catching on its balance between precision and recall, which would eventually prove strong enough to capture both local patterns and word substructures.

**LSTM without FastText.**  This model demonstrates its ability to capture local dependencies and patterns in large text data using recurrent layers, achieving an accuracy of 98.33%, the peak accuracy achieved by this LSTM model. The patterns that undergo such changes, such as shifts in sentiment, are extremely important for their performance in fake detection, as those often involve imperceptible changes in textual properties.

**LR with TF-IDF.**  LR proves less effective than DL models, despite achieving an accuracy of 98.79%. It is both interpretive and computationally efficient, but it also has a sparse feature representation that hinders its ability to capture complex semantic and contextual relationships in textual data. Hence, constructing such an appropriate model becomes very challenging for recognizing subtle patterns hidden in the content of fake news, which typically requires sophisticated modeling techniques to capture fine word-level and contextual characteristics.

**Results with proposed GloVe with CNN+BiLSTM.**  The highest score was also obtained using GloVe with the combination of CNN and BiLSTM processors, making the accuracy yield 99.63%. The reason is that the Bi-directional LSTM extracts sequences of dependency relations and relationships that have been semantically captured. It means that GloVe would collect more knowledge and input on relations and data for world associations and global word relationships. This model achieved the highest precision and recall, making it the top model for distinguishing between fake news and real news, with multiple levels of text understanding.

## Class-wise performance

To further evaluate the models' performance, the class-wise precision, recall, and F1-scores are analyzed in Table 8, which highlights the metrics for each class (Fake and Real news).

**Table 7. The model performance comparison.**

| Model | Accuracy |
|---|---|
| BiLSTM + FastText | 95.53% |
| LSTM w/o FastText | 98.33% |
| LR w TF-IDF | 98.79% |
| Glove with CNN-BiLSTM | 99.63% |

**Table 8**. The class-wise performance analysis.

| Model | Class | Precision | Recall | F1-Score | ROC-AUC |
|---|---|---|---|---|---|
| FastText + BiLSTM | Fake News (0) | 0.96 | 0.95 | 0.96 | 0.9710 |
| | Real News (1) | 0.95 | 0.96 | 0.95 | |
| LSTM w/o FastText | Fake News (0) | 0.98 | 0.99 | 0.98 | 0.9710 |
| | Real News (1) | 0.99 | 0.98 | 0.98 | |
| LR w TF-IDF | Fake News (0) | 0.99 | 0.99 | 0.99 | 0.9989 |
| | Real News (1) | 0.99 | 0.99 | 0.99 | |
| GloVe + CNN-BiLSTM | Fake News (0) | 1.00 | 0.99 | 1.00 | 0.9989 |
| | Real News (1) | 0.99 | 1.00 | 1.00 | |

**Interpretation.**   The comparative performance indicates that the GloVe containing CNN-BiLSTM is the one with the best score of 99.63%. This shows, indeed, that the most effective model for detecting false news can identify both local and global text patterns. It's not the same thing as saying that ML based on LR with TF-IDF is an amenable model with an accuracy of 98.79%. Using sparse feature representations limits its ability to deal with complex semantic nuances. It works just fine by giving 98.33% accuracy without FastText, as an LSTM method performs better in capturing local dependencies of text data. BiLSTM effectively captures meaning relationships within terms, but its accuracy is relatively low at 95.53%, indicating room for improvement compared to models utilizing richer embeddings and hybrid architectures.

## Explainability with LIME

A significant strength of this study is the application of Explainable AI techniques, such as LIME, to interpret the predictions made by the models. LIME explains a model's predictions as approximations in a neighbourhood of a given prediction using an interpretable model. After training and evaluating the models (LSTM, GloVe + BiLSTM, FastText + BiLSTM, and LR with TF-IDF), we applied LIME to understand the key features influencing the predictions made by the models. Instead of explaining individual models, we used LIME to interpret the predictions on the final test dataset, as shown in Figs 8 and 9.

The LIME analysis in Fig 8 indicates that the article is correctly labelled as Real, with a confidence score of 1.00. The words "Reuters", "Trover", "economy", and "stalemate" are flagged by LIME as the primary factors contributing to the prediction. They appear in political, financial, and official settings, which are key parts of real journalism.

The LIME analysis in Fig 9 shows how the audience understands a fake news article about the previous U.S. President Donald Trump's politically charged remarks. According to the model, the article is Fake with a probability of 100 percent. LIME brings out the words "NEVER", "traitor", "prison", and "piece", pointing out that this type of language tends to be loaded with emotion, offers opinions, and uses strong words that are typical in misinformation or editorials.

In addition, Explainable AI methods, such as LIME, can help make sense of the decisions made by these models. This involves identifying the keywords and semantic relationships that drive the model's predictions. Hence, it provides insight into fake news from the model's perspective. Therefore, it fosters transparency, which is essential for gaining trust in the systems that detect fake news, especially in real-world applications where accountability is in question.

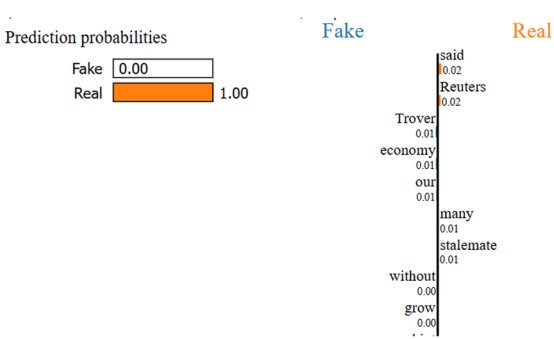

**Fig 8. The explainability analysis for fake class.**

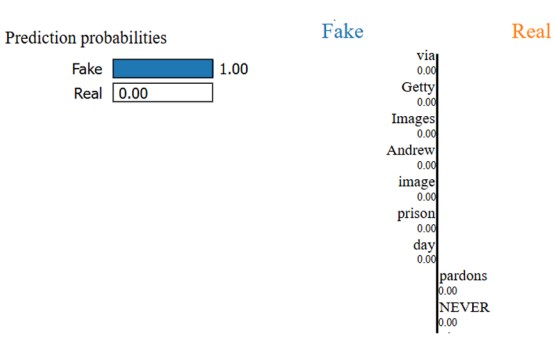

**Fig 9. The explainability analysis for real class.**

## State of the art comparison

The GloVe-Based CNN-Bidirectional LSTM Model has achieved the highest accuracy of 99.63% in fake news detection, which surpasses all current state-of-the-art methods. When compared to FakeBERT (98.90%), BiLSTM with self-attention layers (98.65%), and Multitask learning with CapsNet and BiLSTM (97.96%), it is quite evident that this model outperforms its predecessors in detecting fake news. Another recent model, AugFake-BERT, achieved 92.45% accuracy for fake news classification. Table 9 shows that the success of this model is due to the impactful feature representation provided by the powerful GloVe embedding, along with the robust CNN and BiLSTM architectures, which can capture both the semantic and contextual dependencies present in the text. The results confirmed the power of the

**Table 9. State of the art comparison.**

| Ref | Proposed Technique | Accuracy |
|---|---|---|
| [14] | BERT + CNN for fake news detection | 98.90% |
| [16] | BiLSTM with self-attention layers | 98.65% |
| [17] | Multitask learning with CapsNet and BiLSTM | 97.96% |
| [33] | AugFake-BERT | 92.45% |
| [34] | Courtroom-FND | 78.60% |
| [35] | CAPE-FND | 81.20% |
| **Proposed** | **Glove Based CNN-BILSTM Model** | **99.63%** |

present model in achieving realistic accuracy and reliability in fake news detection, and thus, it significantly contributes to this field.

An illustration, as shown in Fig 10, through a bar chart, reveals the comparison between existing and proposed modern fake news detection methodologies. The proposed Glove-based CNN-Bilstm model achieves the highest accuracy of 99.63% as it demonstrates superiority over existing methods.

### Error analysis

The proposed model demonstrates high performance, achieving an accuracy above 99%. However, there remains a margin of error of approximately 1%. A detailed error analysis reveals that some misclassified instances include deceptive articles that mimic the tonal and linguistic style of legitimate news sources. These articles often use formal language, maintain an objective tone, and include credible-sounding named entities, which can cause the model to mistakenly classify them as genuine. Such articles may also embed misleading narratives within seemingly factual claims, which even confuse ensemble classifiers. Additionally, some honest articles were incorrectly flagged as fake. These were often conspiracy-related or covered controversial topics, frequently featuring sensational or emotionally charged language, which likely contributed to the misclassification.

Another challenge arises when the model is presented with very short news snippets or headlines that lack sufficient context. Due to the limited text, models—especially those relying on word embeddings- struggle to extract meaningful semantic features, resulting in classification errors.

### Discussions

In the current digital era, fake news, intentionally false information or misinformation intended to sway public opinion, is quickly emerging as one of the main problems. Unfortunately, social media and websites are the main channels of this proliferation. It truly could not have affected life at any other time from sociological to political and financial avenues so quickly and to such an unprecedented degree. When such propaganda is discovered, it must

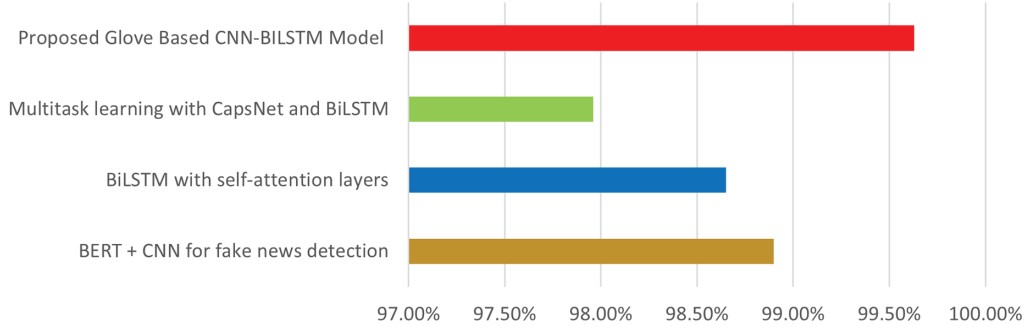

**Comparative analysis of State-of-the-art Fake news detection techniques.**

**Fig 10. Comparative analysis of state-of-the-art fake news detection techniques.**

be stopped to preserve the integrity of information and the democratic process. The primary issue with conventional fake news detection techniques is that they frequently rely on manual fact-checking. It is not adaptable, however, and it requires a significant amount of time. On the other hand, the detection process is automated mainly due to the use of DL and ML models. Such algorithms utilize natural language processing techniques to analyze textual content and identify trends that may be associated with fake news. This study aims to investigate several ML and DL architectures for efficiently classifying news items as authentic or fake. However, due to issues such as domain-specific nuances, the need for generalizability across datasets, and the complexity of deceptive language, identifying fake news remains a challenging endeavor. Explainable AI emphasizes the need for explainability, which boosts the model's prediction and system confidence. For the hybrid model, we include FastText-BILSTM, LSTM, and LR with TF-IDF and GloVe, along with CNN-BiLSTM embedding, to achieve state-of-the-art performance using the explainable AI technique to determine the truthfulness of news. This study demonstrates the effectiveness of these models in accurately identifying false information.

## Conclusion and future directions

This study evaluated various ML and DL models for detecting fake news. The FastText-BiLSTM hybrid model, the LSTM model, and the GloVe-based CNN-BiLSTM model demonstrated strong performance in classifying news articles as either real or fake. By leveraging advanced embeddings provided by FastText and GloVe, these models effectively captured the semantic and contextual relationships within the texts, making the detection of deception more efficient. Additionally, the issue of class imbalance was addressed using SMOTE, which improved the generalization of the developed models. DL techniques have outperformed traditional ML approaches, such as LR, in tasks like fake news detection. This highlights the capability of neural networks to understand and model complex patterns in natural language. The application of explainable AI techniques, such as LIME, further clarified the models' decision-making processes, thereby enhancing transparency and trust in their predictions.

Looking ahead, fake news detection applications will become increasingly valuable as more efficient models utilize advanced techniques, such as attention mechanisms or transformer-based models like BERT or RoBERTa, to grasp language with greater nuance. The hybrid models developed in this study can be improved further through hyperparameter fine-tuning and by incorporating multi-modal data, such as images and videos, to enhance their robustness. Expanding the dataset will also broaden coverage across various topics and timeframes, significantly improving the generalizability of the models across different domains. For practical applications, real-time fake news detection systems should be developed using lightweight or compressed DL models. This approach represents a promising direction for implementing effective detection systems. Various tools, including browser extensions, APIs, and integrations within social media platforms, could be employed to combat the widespread dissemination of misinformation on a large scale.

## Supporting information

**S1 Code. The source code of the experiments.** The source code for the study is provided in the accompanying Jupyter Notebook file.
(IPYNB)

## Author contributions

**Conceptualization:** Sidra Khalid, Shabana Ramzan, Muhammad Munwar Iqbal, Nisrean Thalji, Ali Raza, Aseel Smerat, Changgyun Kim, Norma Latif Fitriyani, Muhammad Syafrudin.

**Data curation:** Sidra Khalid, Norma Latif Fitriyani.

**Formal analysis:** Sidra Khalid, Muhammad Munwar Iqbal, Ali Raza, Norma Latif Fitriyani, Muhammad Syafrudin.

**Funding acquisition:** Changgyun Kim, Muhammad Syafrudin.

**Investigation:** Shabana Ramzan, Muhammad Munwar Iqbal, Nisrean Thalji, Ali Raza, Aseel Smerat.

**Methodology:** Sidra Khalid, Changgyun Kim, Norma Latif Fitriyani, Muhammad Syafrudin.

**Project administration:** Shabana Ramzan, Nisrean Thalji, Muhammad Syafrudin.

**Resources:** Muhammad Munwar Iqbal, Nisrean Thalji, Aseel Smerat.

**Software:** Sidra Khalid, Ali Raza, Norma Latif Fitriyani.

**Supervision:** Shabana Ramzan, Changgyun Kim, Muhammad Syafrudin.

**Validation:** Shabana Ramzan, Muhammad Munwar Iqbal, Ali Raza, Aseel Smerat, Norma Latif Fitriyani.

**Visualization:** Sidra Khalid, Norma Latif Fitriyani.

**Writing – original draft:** Sidra Khalid, Shabana Ramzan, Muhammad Munwar Iqbal, Ali Raza, Norma Latif Fitriyani.

**Writing – review & editing:** Nisrean Thalji, Aseel Smerat, Changgyun Kim, Muhammad Syafrudin.

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
