## [Decision Letter · Decision Letter 0]

9 Apr 2025

PONE-D-25-02599Harnessing Interpretable Novel Combination of GloVe Embedding With Deep CNN-BiLSTM Neural Network for Fake News DetectionPLOS ONE

Dear Dr. Syafrudin,

Thank you for submitting your manuscript to PLOS ONE. After careful consideration, we feel that it has merit but does not fully meet PLOS ONE’s publication criteria as it currently stands. Therefore, we invite you to submit a revised version of the manuscript that addresses the points raised during the review process.

This manuscript is good and interesting, but there still has some significant disadvantage that need to be solve. Please carefully read and solve the two reviews' questions, comments and suggestions. Also, please provide a detailed response letter for each reviewer's comment.

We look forward to receiving your revised manuscript.

Kind regards,

Weiqiang (Albert) Jin, Ph.D.

Academic Editor

PLOS ONE

2. We are unable to open your Supporting Information file [S1 Code.ipynb]. Please kindly revise as necessary and re-upload.

3. Thank you for uploading your study's underlying data set. Unfortunately, the repository you have noted in your Data Availability statement does not qualify as an acceptable data repository according to PLOS's standards.

Additional Editor Comments:

This manuscript is good and interesting, but there still has some significant disadvantage that need to be solve. Please carefully read and solve the two reviews' questions, comments and suggestions. Also, please provide a detailed response letter for each reviewer's comment.

Reviewers' comments:

Reviewer's Responses to Questions

**Comments to the Author**

1. Is the manuscript technically sound, and do the data support the conclusions?

Reviewer #1: Yes

Reviewer #2: Yes

2. Has the statistical analysis been performed appropriately and rigorously? 

Reviewer #1: Yes

Reviewer #2: Yes

3. Have the authors made all data underlying the findings in their manuscript fully available?

Reviewer #1: Yes

Reviewer #2: No

4. Is the manuscript presented in an intelligible fashion and written in standard English?

Reviewer #1: Yes

Reviewer #2: No

5. Review Comments to the Author

Reviewer #1: Thank you for inviting me to review this manuscript. This topic is highly relevant in today's age, where the spread of misinformation poses significant challenges to society. Overall, the research appears to be well-conducted and provides valuable contributions into improving the accuracy of fake news detection.

However, I have several suggestions that I believe could further strengthen the manuscript and enhance its clarity, comprehensiveness. These suggestions are detailed below.

With regard to Section 1, I thought it would be useful to review existing fake news detection techniques in more detail. I feel that a brief summary of existing technologies, including their advantages and disadvantages, would better highlight the need for this study. In addition, although the issue of fake news in the 2016 US election was mentioned, I think it can be added to some new trends in the spread of fake news in recent years, such as the emergence of multimodal fake news, so that readers can better understand the current challenges.

In addition, in the contribution summary, I think the contributions and innovations of this study and other similar studies can be compared in more detail. For example, the uniqueness of this study in terms of model performance and interpretability can be highlighted. In addition, although the use of XAI technology to improve the interpretability of the model is mentioned, I think you can explain the argument further.

In the introduction of relevant work, that is, Section 2, I think a more systematic classification and summary of the references can be made. For example, work can be divided into several categories, such as machine learning-based methods, deep learning-based methods, and interpretable AI-based methods, and then a brief summary of each category. Also, if you think it's reasonable, I recommend quoting the following jobs that do fake news in the Related Work section: 1. A prompting multi-task learning-based veracity dissemination consistency reasoning augmentation for few-shot fake news detection 2. Fake News Detection on Social Media: A Data Mining Perspective 3. Veracity-Oriented Context-Aware Large Language Models–Based Prompting Optimization for Fake News Detection

In the data preprocessing Section in Section 3, I noticed that the author used SMOTE to solve the data imbalance problem. I would like to ask why SMOTE was chosen over other methods, such as undersampling? I think it would be more convincing if I explained the reasons for my choice. Also, in the feature embeddings section, although FastText and GloVe embeddings are mentioned, I thought it would be easier for the reader to add a flow chart showing how the text is converted into embedded vectors.

In addition, I think the data set description in Chapter 3 can be supplemented with the sources and collection methods of data sets. This will make the reader more aware of the quality and representativeness of the data. As for the data preprocessing part, you can also discuss the specific preprocessing steps in depth.

In Section 4, in the experimental design section, you chose an 80%-20% data segmentation ratio. I would like to know why this ratio is chosen instead of other ratios, such as 70%-30%? I think it would be better to add the reasons for choosing this ratio. In addition, I suggest that in the model evaluation section, I think it is possible to add an explanation of the meaning and importance of indicators such as accuracy, accuracy, recall and F1 scores.

In Section 5, the Future Work section, I feel I can describe in more detail the specific application of the advanced models and techniques mentioned. For example, discuss how these models and techniques can be applied to fake news detection and their potential impact.

Overall, the manuscript presents a thorough investigation into leveraging advanced deep learning models and explainable AI techniques for FND area. Given the importance of the topic and the potential impact of the research, I recommend a Major Revision to address the detailed suggestions provided in this review.

Reviewer #2: Title: Harnessing Interpretable Novel Combination of GloVe Embedding with Deep CNN-BiLSTM Neural Network for Fake News Detection

ID: PONE-D-25-02599

I congratulate the authors for their contribution and hard work. Before going to publish the manuscript, a few perhaps would get clarified from my end.

1. In this work, the authors proposed comparative analysis of innovative hybrid deep learning models and embedding techniques-focusing interpretability using Xplainable Artificial Intelligence (XAI) for fake news detection. Elaborate, how it is better compared to the existing methods. Provide more clarifications.

2. Add more recent references and compare proposed method with Existing works.

3. The paper is technically sound but need to improve the grammar.

4. Provide more relevant information regarding the figures.

5. The alignment of data available in figures 7 and 8 are not visible to identify the fake news.

6. Polish the Language.

7. Instead of Tables, Compare the Existing and proposed methods using Bar Graphs to prove proposed method is more Superior.

9. Problem statement is not clearly mentioned.

10. Related work needs to be improved.

11. The Sections are not aligned properly.

6. PLOS authors have the option to publish the peer review history of their article (what does this mean?). If published, this will include your full peer review and any attached files.

Reviewer #1: No

Reviewer #2: No

---

## [Author Response · Author response to Decision Letter 1]

14 May 2025

Response to Reviewers Comments

Manuscript Number: PONE-D-25-02599

Journal: PLOS ONE

Title: Harnessing Interpretable Novel Combination of GloVe Embedding With Deep CNN-BiLSTM Neural Network for Fake News Detection

Dear Editor,

Thank you very much for allowing a resubmission of our manuscript, with an opportunity to address the reviewers’ comments.

We would like to thank the editor and all the worthy reviewers for their efforts, valuable comments and suggestions. The comments given improved the quality of our paper. Based on the feedback, we have extensively revised our manuscript. The updated manuscript with highlighting indicates changes. The detailed modifications to address reviewers’ comments are provided point-by-point in the following.

(Khalid et al.)

Reviewer Comments to Author

Reviewer: 1

Thank you for inviting me to review this manuscript. This topic is highly relevant in today's age, where the spread of misinformation poses significant challenges to society. Overall, the research appears to be well conducted and provides valuable contributions into improving the accuracy of fake news detection.

However, I have several suggestions that I believe could further strengthen the manuscript and enhance its clarity, comprehensiveness. These suggestions are detailed below.

Author response:

The authors are highly grateful for your efforts and insightful comments. We apologize for the inconvenience and inappropriate language that raised ambiguity. The paper is extensively revised to remove ambiguity.

Point 1: With regard to Section 1, I thought it would be useful to review existing fake news detection techniques in more detail. I feel that a brief summary of existing technologies, including their advantages and disadvantages, would better highlight the need for this study. In addition, although the issue of fake news in the 2016 US election was mentioned, I think it can be added to some new trends in the spread of fake news in recent years, such as the emergence of multimodal fake news, so that readers can better understand the current challenges. In addition, in the contribution summary, I think the contributions and innovations of this study and other similar studies can be compared in more detail. For example, the uniqueness of this study in terms of model performance and interpretability can be highlighted. In addition, although the use of XAI technology to improve the interpretability of the model is mentioned, I think you can explain the argument further.

Author response: As per your valuable suggestion, we have added a brief summary of existing technologies, including their advantages and disadvantages, would better highlight the need for this study the introduction section in the updated version of the manuscript as:

“Researchers have explored many strategies [9] in recent times for the detection of fake news among traditional ML models. ML practitioners use conventional algorithms Support Vector Machines (SVM), with the help of feature extraction tools, TFIDF, and Bag-of-Words. Such detection methods are easily implementable but provide limited engineering features and lack the necessary capacity to understand contextual and semantic linguistic meaningfulness. Contextual modeling happens automatically in deep learning models, which include Convolutional Neural Network(CNN) and Recurrent Neural Networks(RNN), together with transformer models BERT and RoBERTa that achieve superior performance outcomes. Recent studies [10] have investigated combination methods between textual content and visual components and contextual factors because these approaches show an improved ability to discover fake news on multiple platforms. GloVe embeddings provide a strong framework by partnering with deep CNN-BiLSTM networks to obtain the best attributes from the word context sense, together with deep neural network capability. The deployment of these methods produces successful extraction of syntactic and semantic elements while providing an interpretable model that addresses current difficulties with most DL systems. The existing detection methods face restrictions that prevent them from supporting generalization in different scenarios, handling multilingual content and scarce resources, and identifying weak or altered material.”

As per your valuable suggestion, we have added to some new trends in the spread of fake news in recent years the introduction section in the updated version of the manuscript as:

“Fake news proliferation through the internet has reached a critical status because it was exposed during the 2016 U.S. presidential election when misinformation demonstrated its capability to sway massive opinion shifts [2]. After the 2016 U.S. presidential election, fake information transmission kept maturing through COVID-19 vaccine conspiracies during the Russia-Ukraine conflict, and deceptive disaster accounts about natural disasters. Multimodal fake news that combines text with images and audio and video elements has become a widespread form of deceptive information since its emergence in recent years. This new version of fake news presents a more compelling, deceptive framework while simultaneously becoming harder for audiences to detect. The complex environment of fake news detection has become even more difficult to manage because of deepfakes alongside other AI-generated media, which makes existing detection systems less effective.”

As per your valuable suggestion, we have enhanced the contribution summary in the introduction section in the updated version of the manuscript as:

“Research contributions

The key contributions of this research are:

The research explores many advanced fake news detection models through FastText-BiLSTM and GloVe with CNN-BiLSTM, LSTM without FastText, and Logistic Regression with TF-IDF, which are benchmarked using various datasets for identifying their best and worst attributes.

The GloVe with CNN-BiLSTM model exhibits excellent state-of-the-art capability that surpasses current standards for fake news detection by achieving high precision with robust performance.

The integration of XAI methodology incorporates LIME (Local Interpretable Model-agnostic Explanations) technology to make CNN-BiLSTM decision processes more understandable, which increases trust in prediction outcomes.

The main innovative aspect of this research develops a framework that maintains excellent detection accuracy and enhances interpretable models for addressing dual performance and interpretability requirements in fake news detection systems.

The approach in this research combines the best features of accuracy and interpretability, while other studies have usually focused on one aspect. The existing deep learning approaches for fake news detection achieve high accuracy metrics at the cost of interpretability. This research distinguishes itself by combining LIME with GloVe and CNN-BiLSTM to deliver high performance with interpretability in fake news detection systems, which aims to establish both effective and trustworthy systems.

LIME enhances the transparency of CNN-BiLSTM by showing the predictive features while performing its operations within the model. The detection system demands transparency to create trust when applied in genuine fake news scenarios.”

As per your valuable suggestion, we have enhanced the XAI technology summary in the introduction section in the updated version of the manuscript as:

“Despite significant advancements in fake news detection using deep learning and NLP, current systems often prioritize accuracy at the expense of interpretability. Moreover, many existing approaches lack the ability to generalize across different languages and platforms. The growing complexity of fake news, particularly through multimodal and AI-generated content such as deepfakes, has further challenged the effectiveness of traditional models. These models typically fail to offer transparency, which is crucial for building user trust in real-world applications.”

Point 2. In the introduction of relevant work, that is, Section 2, I think a more systematic classification and summary of the references can be made. For example, work can be divided into several categories, such as machine learning-based methods, deep learning-based methods, and interpretable AI-based methods, and then a brief summary of each category. Also, if you think it's reasonable, I recommend quoting the following jobs that do fake news in the Related Work section: 1. A prompting multi-task learning-based veracity dissemination consistency reasoning augmentation for few-shot fake news detection 2. Fake News Detection on social media: A Data Mining Perspective 3. Veracity-Oriented Context-Aware Large Language Models–Based Prompting Optimization for Fake News Detection

Author response:

As per your valuable suggestion, we have divided the work into several categories in the literature section in the updated version of the manuscript.

In addition, we have discussed your provided related work in in the updated version of the manuscript as:

Point 3. In the data preprocessing Section in Section 3, I noticed that the author used SMOTE to solve the data imbalance problem. I would like to ask why SMOTE was chosen over other methods, such as undersampling? I think it would be more convincing if I explained the reasons for my choice. Also, in the feature embeddings section, although FastText and GloVe embeddings are mentioned, I thought it would be easier for the reader to add a flow chart showing how the text is converted into embedded vectors.

Author response:

As per your valuable suggestion, we have added the reasons for using SMOTE in the Data Balancing section in the updated version of the manuscript as:

“In Contrast with under sampling involves random minority sample removal that could waste important data points, while SMOTE enhances class balance by conserving original samples. Through this technique, the model can acquire equivalent information from the classes without losing important samples. We chose SMOTE over other oversampling methods due to the moderate imbalance in our dataset, as it allowed us to preserve the diversity of training samples.”

In addition, we have added a flowchart in the Feature Embedding Representations section in the updated version of the manuscript as:

“The embedding representation process is illustrated in full detail through Figure 3. This figure demonstrates the tokenization steps that generate numerical input through the pre-trained word embeddings of FastText and GloVe. The diagram demonstrates a method to handle out-of-vocabulary words through zero vector assignment, which provides consistent coverage for every token.”

Fig 3. Flowchart of the embedding process converting news text into vectorized input using FastText or GloVe for deep learning models.

Point 4. In addition, I think the data set description in Chapter 3 can be supplemented with the sources and collection methods of data sets. This will make the reader more aware of the quality and representativeness of the data. As for the data preprocessing part, you can also discuss the specific preprocessing steps in depth.

Author response:

As per your valuable suggestion, we have revised the dataset description in the updated version of the manuscript as:

“The real news articles were gathered from the Reuters news agency, a globally recognized and credible source for authentic journalism. The information collection process combined automated scraping techniques for sorting articles from these sources with manual verification operations to validate record authenticity. These scholarly articles were published across the years from 2016 to 2017 during which fake news distribution reached its peak especially in political environments.”

Furthermore, we have expanded the data preprocessing section to discuss the specific preprocessing steps in greater depth in the updated version of the manuscript as:

“Data Preprocessing

Data preprocessing is crucial in order to prepare the raw text data for embedding generation and input to machine learning models. The following detailed steps outline the preprocessing methods applied to the dataset:

Tokenization is done manually, without help from external libraries. It included breaking a text column down to single words (tokens) by whitespace and punctuation. Tokenization makes it a structured textual representation of raw text broken down into smaller analyzable units.

After tokenization, the sequences of tokens had different lengths. To maintain uniformity and help gain compatibility with deep learning models, all sequences were padded to a fixed length of 500 tokens. This was done with the pad sequences method, which adds zeroes in front of shorter sequences. The maximum sequence length (500 tokens) was chosen based on the longest article in the dataset to ensure that not a single bite of meaningful information was lost up to truncation.”

Point 5. In Section 4, in the experimental design section, you chose an 80%-20% data segmentation ratio. I would like to know why this ratio is chosen instead of other ratios, such as 70%-30%? I think it would be better to add the reasons for choosing this ratio. In addition, I suggest that in the model evaluation section, I think it is possible to add an explanation of the meaning and importance of indicators such as accuracy, accuracy, recall and F1 scores.

Author response:

As per your valuable suggestion, we have added the reasons 80%-20% in the experimental design section in the updated version of the manuscript as:

“The dataset followed stratified sampling to divide itself into (80\%) training data and (20\%) testing data. These particular ratios were selected to achieve adequate training data quantity along with a reliable, unbiased evaluation subset. Using a split of (80\%)-(20\%) in machine learning experiments is standard practice when working with datasets containing 44,000 records and above, since it provides optimal training and testing performance. A (20\%) test set was utilized throughout the experiment, running on Google Colab, because it decreased resource usage and computation time, yet upheld evaluation standards.”

In addition, we have added an explanation of accuracy, recall and F1 scores metrics in the updated version of the manuscript as:

“”

Point 6. In Section 5, the Future Work section, I feel I can describe in more detail the specific application of the advanced models and techniques mentioned. For example, discuss how these models and techniques can be applied to fake news detection and their potential impact.

Author response:

As per your valuable suggestion, we have revised the Future Work in the updated version of the manuscript as:

“Future Work

In the future, fake news detection applications will prove valuable when more efficient models use advanced techniques, such as attention mechanisms, or transformer-based models like BERT or RoBERTa, to understand language with very sophisticated nuances. The hybrid models developed in this work can be taken towards further perfection by hyperparameter fine-tuning and including multi-modal data, such as images or videos, to improve robustness. Increasing the dataset will also extend definition coverage on topics and timeframes, vastly improving generalizability with models across different domains. Real-time fake news detection systems need to be deployed through the use of lightweight or compressed deep learning models as a valuable direction for implementing the detection systems. A range of tools, including browser extensions, APIs, and integration within social media platforms, could serve as deployments to stop the spread of misinformation on a large scale.”

Point 7. Overall, the manuscript presents a thorough investigation into leveraging advanced deep learning models and explainable AI techniques for FND area. Given the importance of the topic and the potential impact of the research, I recommend a Major Revision to address the detailed suggestions provided in this review.

Author response:

We again apologize for the inconvenience and inappropriate language that raised ambiguity. The updated paper is extensively revised to remove ambiguity.

Reviewer: 2

I congratulate the authors for the

---

## [Decision Letter · Decision Letter 1]

9 Jun 2025

PONE-D-25-02599R1Harnessing Interpretable Novel Combination of GloVe Embedding With Deep CNN-BiLSTM Neural Network for Fake News DetectionPLOS ONE

Dear Dr. Syafrudin,

Thank you for submitting your manuscript to PLOS ONE. After careful consideration, we feel that it has merit but does not fully meet PLOS ONE’s publication criteria as it currently stands. Therefore, we invite you to submit a revised version of the manuscript that addresses the points raised during the review process.

**ACADEMIC EDITOR: **The two anonymous reviewers have proposed several suggestions and both of them think you should proof your language skills and your writing to make your paper easy-following. So please complete and solve their concerns in the next two months.

We look forward to receiving your revised manuscript.

Kind regards,

Weiqiang (Albert) Jin, Ph.D.

Academic Editor

PLOS ONE

Journal Requirements:

Additional Editor Comments:

The two anonymous reviewers have proposed several suggestions and both of them think you should proof your language skills and your writing to make your paper easy-following. So please complete and solve their concerns in the next two months.

Reviewers' comments:

Reviewer's Responses to Questions

**Comments to the Author**

1. If the authors have adequately addressed your comments raised in a previous round of review and you feel that this manuscript is now acceptable for publication, you may indicate that here to bypass the “Comments to the Author” section, enter your conflict of interest statement in the “Confidential to Editor” section, and submit your "Accept" recommendation.

Reviewer #1: (No Response)

Reviewer #2: All comments have been addressed

2. Is the manuscript technically sound, and do the data support the conclusions?

Reviewer #1: Yes

Reviewer #2: Yes

3. Has the statistical analysis been performed appropriately and rigorously? 

Reviewer #1: Yes

Reviewer #2: Yes

4. Have the authors made all data underlying the findings in their manuscript fully available?

Reviewer #1: No

Reviewer #2: Yes

5. Is the manuscript presented in an intelligible fashion and written in standard English?

Reviewer #1: No

Reviewer #2: No

6. Review Comments to the Author

Reviewer #1: Although this paper has been reviewed by the other reviewer in the last round, but I also found some issues that the author should pay attention to, which are listed as follows:

1. The state-of-the-art comparison (Table 8) needs updating with more recent works (2023-2024) to properly contextualize the claimed novelty. The field evolves rapidly.

2. The LIME explanations (Figures 8-9) require much more detailed interpretation. What do the highlighted words indicate about the model's decision process? How representative are these examples?

3. The dataset section should address potential source biases (e.g., PolitiFact's political leanings) and how this might affect model generalizability.

4. The methodology needs clearer justification for architectural choices (e.g., why 128 units in BiLSTM? Why 5 epochs?) and hyperparameter tuning process.

5. Moreover, I suggest the author cite the following literature and work in the Related Work section: 1. Fake news detection: comparative evaluation of BERT-like models and large language models with generative AI-annotated data https://arxiv.org/abs/2412.14276 2. Veracity‐Oriented Context‐Aware Large Language Models–Based Prompting Optimization for Fake News Detection 3. Courtroom-FND: a multi-role fake news detection method based on argument switching-based courtroom debate 4. A prompting multi-task learning-based veracity dissemination consistency reasoning augmentation for few-shot fake news detection.

6. An error analysis section should be added - what types of fake news still evade detection? Where do the models consistently fail?

7. The model architectures lack sufficient detail about hyperparameter selection and tuning processes. Please introduce the key hyper-parameters in the update experiments section.

8. Moreover, I find that the important metrics like ROC-AUC are missing, which are particularly valuable for evaluating performance on imbalanced data.

9. The language needs significant polishing throughout - many awkward phrasings and grammatical errors. I suggest the authors should make some language proof serives to make their paper become more easy-understanding.

Reviewer #2: Respected Editor,

The authors have addressed all the Comments thoroughly except the Language. Try to Polish the Language.

7. PLOS authors have the option to publish the peer review history of their article (what does this mean?). If published, this will include your full peer review and any attached files.

Reviewer #1: **Yes: **Ningwei Wang

Reviewer #2: No

---

## [Author Response · Author response to Decision Letter 2]

18 Jun 2025

Please see 'Response to Reviewers' file type.

---

## [Decision Letter · Decision Letter 2]

29 Jul 2025

Harnessing Interpretable Novel Combination of GloVe Embedding With Deep CNN-BiLSTM Neural Network for Fake News Detection

PONE-D-25-02599R2

Dear Dr. Muhammad Syafrudin,

We’re pleased to inform you that your manuscript has been judged scientifically suitable for publication and will be formally accepted for publication once it meets all outstanding technical requirements.

Kind regards,

Weiqiang (Albert) Jin, Ph.D.

Academic Editor

PLOS ONE

Additional Editor Comments (optional):

Congratulations, all the reviewers believe that your manuscript has met the publication standards of PLOS ONE. Additionally, before final publication, we recommend carefully reviewing the overall formatting of the article to ensure compliance with the journal's official standards, as well as verifying the accuracy of the reference format.

Reviewers' comments:

Reviewer's Responses to Questions

**Comments to the Author**

1. If the authors have adequately addressed your comments raised in a previous round of review and you feel that this manuscript is now acceptable for publication, you may indicate that here to bypass the “Comments to the Author” section, enter your conflict of interest statement in the “Confidential to Editor” section, and submit your "Accept" recommendation.

Reviewer #1: All comments have been addressed

Reviewer #2: All comments have been addressed

2. Is the manuscript technically sound, and do the data support the conclusions?

Reviewer #1: Yes

Reviewer #2: Yes

3. Has the statistical analysis been performed appropriately and rigorously? 

Reviewer #1: Yes

Reviewer #2: Yes

4. Have the authors made all data underlying the findings in their manuscript fully available?

Reviewer #1: Yes

Reviewer #2: Yes

5. Is the manuscript presented in an intelligible fashion and written in standard English?

Reviewer #1: Yes

Reviewer #2: Yes

6. Review Comments to the Author

Reviewer #1: The authors have addressed all my concerns thoroughly in the revised manuscript, and I find the current version satisfactory for publication. I recommend acceptance in its present form. But the author should pay attention to the conclusion part, because the future direction and the future work part is mutual-repeated. I suggestion to remove the small title "future work". And make the conclusion section become more concise.

Reviewer #2: Dear Authors,

The Manuscript was revised at the extent. I congratulate the authors for their hard work and contributions.

7. PLOS authors have the option to publish the peer review history of their article (what does this mean?). If published, this will include your full peer review and any attached files.

Reviewer #1: **Yes: **Ningwei Wang

Reviewer #2: No

---

## [Editor Report · Acceptance letter]

PONE-D-25-02599R2

PLOS ONE

Dear Dr. Syafrudin,

I'm pleased to inform you that your manuscript has been deemed suitable for publication in PLOS ONE. Congratulations! Your manuscript is now being handed over to our production team.

Kind regards,

on behalf of

Dr. Weiqiang (Albert) Jin

Academic Editor

PLOS ONE